# Fast Decision Algorithm of CU Size for HEVC Intra-Prediction Based on a Kernel Fuzzy SVM Classifier

Shuqian He [1], Zhengjie Deng [1,*] and Chun Shi [2]

[1] School of Information Science and Technology, Hainan Normal University, Haikou 571158, China
[2] School of Electronic and Information, Guangdong Polytechnic Normal University, Guangzhou 510640, China
* Correspondence: hsdengzj@163.com

**Abstract:** High Efficiency Video Coding (HEVC) achieves a significant improvement in compression efficiency at the cost of extremely high computational complexity. Therefore, large-scale and wide deployment applications, especially mobile real-time video applications under low-latency and power-constrained conditions, are more challenging. In order to solve the above problems, a fast decision method for intra-coding unit size based on a new fuzzy support vector machine classifier is proposed in this paper. The relationship between the depth levels of coding units is accurately expressed by defining the cost evaluation criteria of texture and non-texture rate-distortion cost. The fuzzy support vector machine is improved by using the information entropy measure to solve the negative impact of data noise and the outliers problem. The proposed method includes three stages: the optimal coded depth level "0" early decision, coding unit depth early skip, and optimal coding unit early terminate. In order to further improve the rate-distortion complexity optimization performance, more feature vectors are introduced, including features such as space complexity, the relationship between coding unit depths, and rate-distortion cost. The experimental results showed that, compared with the HEVC reference test model HM16.5, the proposed algorithm can reduce the encoding time of various test video sequences by more than 53.24% on average, while the Bjontegaard Delta Bit Rate (BDBR) only increases by 0.82%. In addition, the proposed algorithm is better than the existing algorithms in terms of comprehensively reducing the computational complexity and maintaining the rate-distortion performance.

**Keywords:** HEVC; intra-coding; CU size decision; fuzzy SVM; rate distortion

## 1. Introduction

Recently, with the development of technology, in order to meet the compression requirements of high-definition and ultra-high-definition video, research on how to further improve the coding efficiency of next-generation video codecs has been carried out on a large scale. ISO/IEC (MPEG) and ITU-T (VCEG) have formed the Joint Collaborative Team on Video Coding (JCT-VC) to develop a next generation video coding standard, which is called High Efficiency Video Coding (HEVC) [1–3]. In recent years, with the promotion of 5G networks and the popularization of mobile smart terminal devices, video data services in video applications such as online education, video conferencing, online games, and video social interaction have exploded. Especially with the emergence of high-definition and ultra-high-definition video services, end users are increasingly demanding high frame rates, a high resolution, and a high dynamic range for real-time video interaction, which brings new challenges to the implementation of video coding. The latest report from Sandvin, an Internet smart solution provider, shows that in 2019, video traffic accounted for more than 60% of the global Internet downlink traffic [1]. The widely used H.264/AVC video coding standard can no longer meet the current real-time video coding needs. Compared with H.264/AVC, the video coding standard (H.265/HEVC) keeps the visual quality unchanged and the transmission bit rate is reduced by 50%, but the computational complexity is high.

To improve the coding rate-distortion performance, the HEVC computational complexity evaluation is increased by nearly three times [4].

Accordingly, many coding efficiency improvement techniques, such as coding tree unit (CTU) coding (from 64 × 64 down to 8 × 8 Luma samples), prediction unit (PU) coding, transform unit (TU) tree coding, uniform intra-prediction and so on, are available in the current HEVC. More details can refer to [1,2,4]. Each picture frame in HEVC is partitioned into many square-shaped CTUs. A CTU represents the most basic processing unit, and each CTU can be recursively split into a smaller CU based on a quadtree structure. The CU sizes range from 8 × 8 up luma samples to 64 × 64. The coding depth order for CUs is referred to as the z-scan and is illustrated for an example in Figure 1.

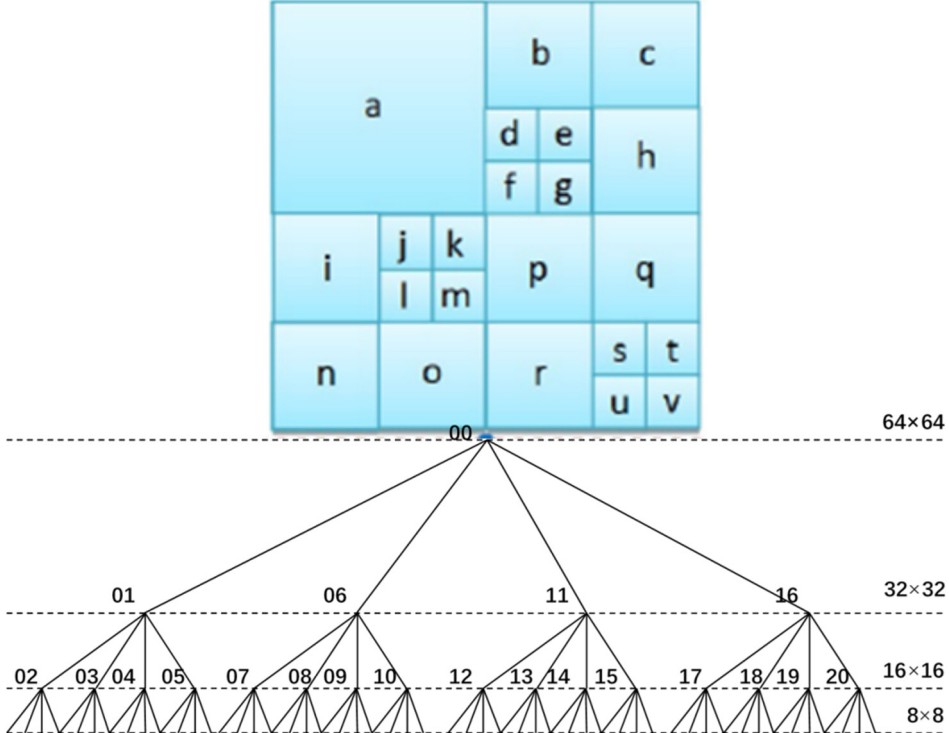

**Figure 1.** Partition of CU and its corresponding coding tree structure.

Due to the extended prediction and transform block size with a flexible coding structure, HEVC can meet the coding requirements of general video sequences and can effectively code video sequences from low to high picture resolutions. Because up to 35 prediction modes are used in HEVC (e.g., mode 0 is used for PLANAR, mode 1 is used for DC and other angle modes) while only 9 modes are used in H.264/AVC, HEVC uses a more complex structure and has more accurate prediction accuracy, and its intra-frame prediction is significantly improved compared to H.264/AVC [3]. The significant increased number of angular intra-prediction modes in HEVC leads to a much higher compression efficiency. However, it also brings a tremendous computational burden to the mode decision process when the expensive rate-distortion optimization (RDO) process is used to search for the minimum rate distortion (RD) cost mode. Therefore, in the real implementation of HEVC codec, the main objective of the fast mode decision method is to reduce the computational complexity while maintaining high video coding RD performance.

Recently, many researchers devoted their efforts to the reduction in the complexity of HEVC for prediction coding. The main idea is to skip or terminate the non-optimal rate-distortion optimization mode selection processing in advance through prediction, avoiding the exhaustive search of RDO for coding unit depth size decision, prediction mode selection, and transform coding unit selection. The choice of decision-making strategies and thresholds is mainly determined by the temporal and spatial statistical characteristics

of the video content and machine learning methods. The following two categories are introduced, respectively:

The method based on spatio-temporal statistical characteristics is mainly completed by manually extracting statistical features, including the spatio-temporal correlation video frames, the correlation of coding unit depth and prediction mode, the correlation of rate distortion, the correlation of content complexity, etc. [5–12]. According to the RD cost of each coding unit CU and the inter-mode prediction error, Shen et al. [5] used the Bayesian decision method to determine the coding unit CU partition in advance [6]. The gradient-based method reduces the range of candidate modes and uses the statistical characteristics of depth difference and least Hadamard transform-based costs (HAD costs) to make early decisions of the support vector coding unit. The fast quadtree pruning algorithm in [7] uses the prediction of residual statistics to reduce coding time. Lei et al. [8] distinguished between the natural content coding unit CU and the screen content coding unit CU for the extension of intra-prediction coding in screen content coding. Based on the depth and mode content statistical characteristics of the temporal and spatial adjacent coding units, the optimal prediction mode is predicted, and the quad-tree structure is divided to reduce the complexity of intra-frame coding. Q. Hu et al. [9] modeled the mode and CU size selection decision as a binary classification problem and used Neyman–Pearson's rules to balance the rate-distortion performance loss and the complexity reduction rate. H.S. Kim et al. [10] proposed a fast CU size decision algorithm based on Bayesian decision rules. Xiong, J. et al. [11] used Sum of Absolute Difference (SAD) features to optimize the motion estimation algorithm, and based on the relationship model between the motion compensation rate-distortion cost and SAD, proposed a fast CU decision method. Yang, H. et al. [12] studied the intra-frame mode selection optimization problem of the latest video coding standard H.266/VVC, explored the new block size and coding mode distribution characteristics, and proposed a fast intra-frame coding algorithm including the low-complexity coding tree unit (CTU) structure decision and fast intra-mode selection. M. Jamali and S. Coulombe [13] defined a low-complexity absolute transform difference criterion to predict the rate-distortion cost of intra-frame modes and combined gradient methods to reduce the number of directional candidate modes.

In recent five years, with the widespread application of machine learning methods, methods based on machine learning mainly use content statistical features to complete classification and decision making through machine learning classifiers or non-linear modeling. Compared with statistical methods, its accuracy is higher and it can better respond to changes in video content [14–24]. Ryu and Kang [14] used the random forest method integrated with random decision trees to predict the intra-prediction mode and achieved the reduction in coding complexity by avoiding the complicated RDO search for unnecessary modes. Zhang, Q. et al. [15] proposed a fast CU partition based on a random forest classifier (RFC) model and a fast intra-mode optimization method based on texture region features for H.266/VVC. L. Zhu et al. [16] modeled the selection process of the coding unit CU and prediction unit PU as a hierarchical classification problem, optimized the selection through a binary and multi-class support vector machine (SVM), and combined offline machine learning modes and online machine learning modes for classifiers to achieve a better prediction performance. X. Liu et al. [17] chose representative video content features to characterize more accurate statistical characteristics, enhanced the prediction performance of support vector machines (SVM), and achieved a better rate-distortion performance. In [18], two SVM classifiers using CU size and an RD cost ratio difference as features were proposed to implement the decision of CU splitting and the early termination of CU. L. Zhu et al. [19] proposed a fast-coding unit method based on a fuzzy support vector machine for the rate-distortion-complexity optimization problem. The decision-making process was expressed as a cascaded multi-level classification decision-making problem, which has higher adaptability. Grellert, M. et al. [20] classified the obvious and inconspicuous cases of intra-frame prediction coding unit size decision by stages. Through the different types of feature information obtained, different classifiers are



used for prediction in the threshold selection or they skip the current depth judgment early. The data-driven method further improves the rate distortion-complexity performance. Xu et al. [21] proposed a CU deep decision algorithm based on deep learning to reduce the computational complexity of intra- and inter-coding in HEVC, in which convolutional neural networks (CNN) and long- and short-term memory networks (LSTM) are used to predict the best CU partition. Based on the original work [21], a network pruning method for real-time coding tree unit division was proposed to accelerate the latest deep neural network model [22]. Additionally, based on the weight parameter retention rate, an adaptive network model pruning scheme was designed, and an effective complexity control method was proposed.

Huang et al. [23] proposed a CTU-level complexity control method, which can adaptively adjust the depth processing range of each CTU according to the target complexity. A CTU-level coding complexity statistical model is introduced, which can accurately estimate and allocate the complexity resources of each CTU to achieve the goal of computational complexity control. Huang, C. et al. [24] defined the optimization acceleration characteristics of different modules in the hybrid coding framework based on the optimization of time saving and the rate-distortion performance, and proposed a heuristic model-oriented framework (HMOF). By fusing the advantages of CNN deep learning technology and naive Bayesian prediction, the framework proposes two advanced acceleration algorithms for the CU and PU modules, respectively. In order to adapt to the different characteristics of video content, Li, T. et al. [25] obtained the precision coding unit CU decision model through an online learning method. According to the multi-level and multi-stage complexity allocation, the complexity budget can be reasonably allocated to each coding level. A good compromise between complexity control and rate-distortion (RD) performance is achieved.

In summary, deep neural networks can extract high-dimensional non-linear features and can achieve a very high prediction accuracy through a large number of parameters and multi-layer neural networks; however, they also cause additional high computational complexity and are not suitable for real-time video applications. In contrast, statistics-based heuristic algorithms require very little additional computational overhead, but the sharp drop in the computational complexity is difficult to maintain a high prediction accuracy. Therefore, the SVM-like traditional machine learning algorithm is a compromise between the previous two algorithms. A satisfactory prediction accuracy is achieved, but the overhead is relatively low.

Different from the existing algorithms, we adopt the online method of fuzzy support vector machines to improve the accuracy of the CU decision model by extracting texture features and non-texture features reasonably. In order to further utilize the encoded information of the coding unit, the method adopts a multi-stage update strategy, which can more reasonably reduce the computational complexity of different depth coding units. In addition, a new initial depth range prediction method is proposed to achieve an effective and flexible compromise between RD performance and coding complexity.

The remainder of this paper is organized as follows. Section 2 details the derivation and statistical analyses. Section 3 describes the scheme for multi-stage CU decision and initial depth range prediction. The experimental results and analysis are provided in Section 4 to validate the proposed scheme. Finally, the concluding remarks are summarized in Section 5.

## 2. Derivation and Analysis

### 2.1. Analysis of HEVC Quadtree Coding

As shown in Table 1, we selected three video sequences with different texture characteristics for the intensive experiment. The optimal segmentation structure of the intra-frame coding tree unit (CTU) was obtained by an exhaustive search, and the experimental results can be used to evaluate the statistical distribution characteristics of the optimal coding unit (CU) size. In our experiment, the All-Intra-Main condition was used, the *Qp*s were set to 22, 27, 32, and 37, the RDO was enabled, the LCU size = 64, and the number of coded

frames = 50. The results are summarized in Table 1. The percentages of the CU depth level of "0", "1", "2", and "3" were 19.8%, 33.7%, 19.9%, and 26.6% on average, respectively. For video sequences with a complex texture (such as "BasketballDrill"), the probability of the depth "0" to be selected was relatively low, i.e., 0.6–17%. While for video sequences containing homogeneous content (such as "Kimono"), the probability of the depth level "0" as the best unit was very high, i.e., 25.6–38.3%. Therefore, the optimal CU size can be pre-determined based on the texture property of each CTU. In addition, it is also noted that most CTUs select the first three depth levels (0–2), especially at a high *Qp*s. Take the sequence "BQTerrace" for example: the percentage of the depth level "3" increases more than 23% when the *Qp* is varied from 22 to 37. Therefore, it is appropriate to choose a larger CU as the optimal structure for CUs with similar video content. Correspondingly, a smaller CU size is more suitable for CUs with complex content. Consequently, the intra-prediction on small CU decisions can be adaptively bypassed for homogeneous content without any coding efficiency degradation. The bottom parts of Figure 1 show the skipping process of CU depth, where the unnecessary coding with the CU (marked with dotted lines) is simply bypassed. Similarly, for the complex content, the large CU sizes can also be skipped without affecting the encoding quality.

**Table 1.** Depth level distribution (%).

| Video Sequences | *Qp*s | Depth Level 0 | Depth Level 1 | Depth Level 2 | Depth Level 3 |
|---|---|---|---|---|---|
| BQTerrace (1920 × 1080) | 22 | 15.9 | 15.9 | 15.1 | 53 |
| | 27 | 23.8 | 17.3 | 20.5 | 38.4 |
| | 32 | 24 | 27.1 | 17.5 | 31.4 |
| | 37 | 23 | 22 | 24.9 | 30.1 |
| BasketballDrill (832 × 480) | 22 | 0.6 | 10.1 | 21.6 | 67.7 |
| | 27 | 1.1 | 22.1 | 31.6 | 45.2 |
| | 32 | 8.7 | 33 | 33.4 | 24.9 |
| | 37 | 17 | 38.1 | 28.8 | 16.1 |
| Kimono (1920 × 1080) | 22 | 38.3 | 47 | 10.3 | 4.4 |
| | 27 | 25.6 | 59.6 | 11.5 | 3.3 |
| | 32 | 28.7 | 56.5 | 12.4 | 2.4 |
| | 37 | 30.7 | 56.1 | 11.5 | 1.7 |
| Average | | 19.8 | 33.7 | 19.9 | 26.6 |

　　　Otherwise, one can easily perceive that a large CU-sized CTU is more suitable for a homogeneous region, while a small CU-sized CTU, on the contrary, is more suitable for a region containing a complex object. In these cases, the combination of different CU sizes is expected to be the optimal prediction choice for LCU. In order to verify this observation, the RDO exhaustive mode decision of the HEVC standard reference software was used. By using the different video sequences listed in Table 2, a relatively extensive test experiment was carried out, and the statistical distribution results of the optimal partition structure of each CTU were obtained. The resulting data are shown in Table 2. This study strongly suggests that depth sets be grouped into different complexity classes in terms of the depth range from a Min value to a Max value. The classes that resulted from such depth grouping are established and summarized in Table 3. This study clearly indicates that class C0, C1, C2 with low complexity and class C4, C5, C6 with high complexity are selected as the optimal partition with a high probability. Consequently, only the depths involved in each class are required to be checked.

**Table 2.** Depth range class distribution (%).

| Video Sequences | *Qps* | Depth Range Category of LCU (%) | | | | | | |
|---|---|---|---|---|---|---|---|---|
| | | **C0** | **C1** | **C2** | **C3** | **C4** | **C5** | **C6** |
| BQTerrace (1920 × 1080) | 22 | 1.1 | 1.6 | 5.6 | 0.1 | 38.9 | 51.2 | 1.5 |
| | 27 | 3.6 | 4.5 | 5.4 | 0.1 | 41.5 | 44.5 | 0.4 |
| | 32 | 4.8 | 4.7 | 8.2 | 0.2 | 46.9 | 35.1 | 0.0 |
| | 37 | 10.3 | 6.3 | 10.4 | 0.2 | 49.0 | 23.8 | 0.0 |
| BasketballDrill (832 × 480) | 22 | 5.5 | 18.8 | 7.5 | 0.4 | 42.5 | 25.1 | 0.2 |
| | 27 | 8.0 | 26.7 | 11.6 | 0.2 | 38.8 | 14.7 | 0.0 |
| | 32 | 13.5 | 19.2 | 19.8 | 0.4 | 36.9 | 10.2 | 0.0 |
| | 37 | 22.2 | 19.2 | 18.0 | 0.0 | 32.7 | 7.8 | 0.0 |
| Kimono (1920 × 1080) | 22 | 9.2 | 10.0 | 24.6 | 1.7 | 35.4 | 19.2 | 0.0 |
| | 27 | 17.1 | 20.4 | 12.1 | 0.0 | 32.1 | 18.3 | 0.0 |
| | 32 | 31.7 | 10.4 | 10.8 | 0.0 | 32.1 | 15.0 | 0.0 |
| | 37 | 39.2 | 6.7 | 14.6 | 0.0 | 27.9 | 11.7 | 0.0 |

**Table 3.** The depth sets and their involved depths.

| Depth Range Class | C0 | C1 | C2 | C3 | C4 | C5 | C6 |
|---|---|---|---|---|---|---|---|
| Involved Depths | 0 | 1 | 1, 2 | 2 | 2, 3 | 1, 2, 3 | 3 |

### 2.2. Cost Relationship between CU Sizes

Similar to H.264/AVC, the *RD* optimization technique is performed in HEVC intra-prediction encoding, and the *RD* costs are computed for all the possible CU sizes to find the minimum *RD* cost. The *RD* cost is defined as follows:

$$J_{RD} = SSE(Qp) + \lambda_{Mode}R(Qp) \tag{1}$$

where *SSE* represents the sum of squared difference between the original and its reconstructed CU blocks, respectively, *R* denotes the total number of bits used for CU signaling, *Qp* is the quantization parameter, and $\lambda_{Mode} = 0.85 \times 2^{(Qp-12)/3}$ is the Lagrange multiplier.

In HEVC, the signaling of the side information for non-texture data, which includes quadtree split information for TU and CU, prediction modes for PU, filter parameters for the reconstructed PU, etc., becomes more important due to the adoption of many complex coding tools with more coding parameters compared to the conventional methods in previous video coding standards. In order to obtain the relationship between the non-texture cost and the texture cost, the bit rate *R* of CU was analyzed. The *R* is composed of:

$$R = R_{nontexture} + R_{texture} \tag{2}$$

where $R_{nontexture}$ denotes the total number of bits for the non-texture information, $R_{texture}$ represents the encoding bits for the transform residues which are derived from prediction, transform, quantization, and entropy coding. Substituting (2) into (1), the RDO cost can be represented as follows:

$$J_{RD} = \underbrace{SSE(Qp) + \lambda_{Mode}R_{texture}(Qp)}_{J_{RD_{texture}}} \\ + \underbrace{\lambda_{Mode}R_{nontexture}(Qp)}_{J_{RD_{nontexture}}} \tag{3}$$

where $J_{RD_{texture}}$ and $J_{RD_{nontexture}}$ are the texture cost and non-texture cost, respectively. The non-texture cost of the CUs has a strong relationship with the depth level of CTUs. As the number of the depth levels increases, more non-texture bits are required to mode the presentation correspondingly. Therefore, a fixed relationship can be determined:

$$J_{RD_{nontexture,depth_i}} < J_{RD_{nontexture,depth_{i+1}}} \tag{4}$$

where $depth_i$ denotes the depth level $i$ of the current CU for $i = 0, 1, 2$.

In the new standards, HEVC has introduced many complicated partition CTU and prediction modes, which has led to significant increases in the non-texture signaling bits. Assuming a certain depth level of a sub-block of a given CTU sample, performing different modes of segmentation under this level of sub-block will obtain different prediction and transformation results, and thus will result in different rate distortion costs. When the current depth level block is further sub-divided into multiple sub-blocks, compared with the current level block, the smaller the block can be described more accurately, and a more accurate prediction and transformation accuracy will be obtained; that is, smaller coding prediction residuals will be obtained. The set of supported sub-divisions for the current level block will produce a smaller bit rate and distortion but will increase the bit rate required to transmit more block prediction parameters. According to the observation, the variance of the residual will decrease proportionally with the growth of the depth level, which results in better prediction, i.e., $\sigma^2_{residual,depth_i} > \sigma^2_{residual,depth_{i+1}}$. With the assumption of the Laplace distribution of the residual signal [16], we have: $SSE(\sigma^2_{residual}|Qp) \propto \sigma^2_{residual}$ and $R_{texture}(\sigma^2_{residual}|Qp) \propto \sigma^2_{residual}$. This proportional relation of $SSE$, $R_{texture}$, and $\sigma^2_{residual}$ can be observed in Figure 2. It can be observed that both $SSE$ and $R_{texture}$ are monotonically increasing with $\sigma^2_{residual}$, so their summation should also be a monotonic increasing function.

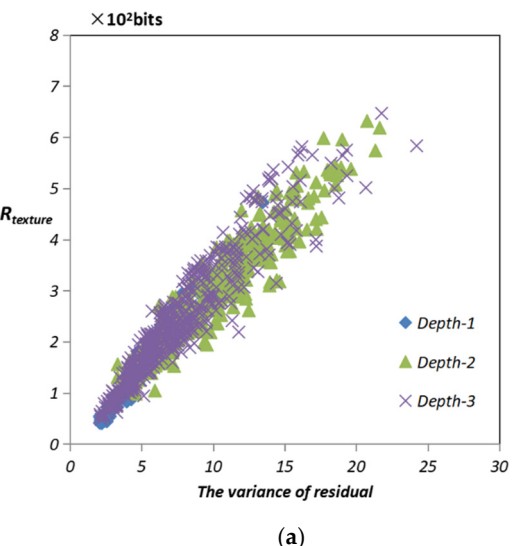

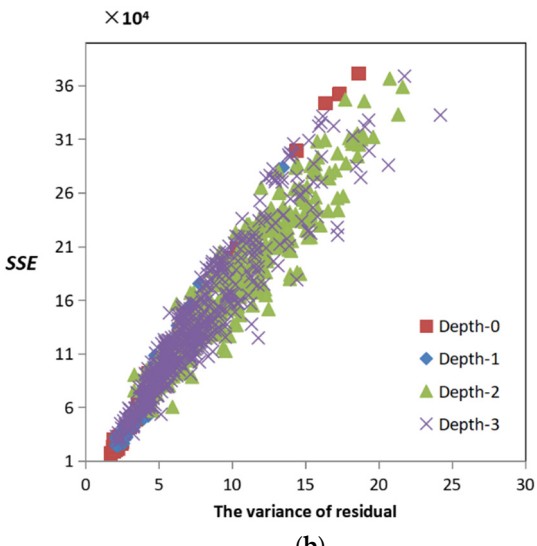

(**a**)                                                                                 (**b**)

**Figure 2.** Relationship between the variance of residual and both *SSE* and rate of CU for Class A sequence "Traffic" (*Qp* = 27).

Therefore, it easily follows that:

$$SSE\left(\sigma^2_{residual}\Big|Qp\right) + \lambda_{Mode}R_{texture}\left(\sigma^2_{residual}\Big|Qp\right) \propto \sigma^2_{residual} \tag{5}$$

and

$$J_{RD_{texture,depth_i}} > J_{RD_{texture,depth_{i+1}}} \tag{6}$$

According to (4) and (6), the relationships of the non-texture rate and texture *RD* cost across successive depth levels can be obtained:

$$\begin{cases} 0 < R_{nontexture,depth_i} < R_{nontexture,depth_{i+1}} \\ J_{RD_{texture,depth_i}} > J_{RD_{texture,depth_{i+1}}} > 0 \end{cases} \tag{7}$$

A measure to evaluate the texture complexity of the current CU depth level is introduced, which is the texture cost percentage of the cost term (*TCPOC*), i.e.,

$$TCPOC_{depth_i} = \frac{J_{RD_{texture_i}}}{J_{RD_i}} \tag{8}$$

Here, $TCPOC_{depth_i}$ can be viewed as an encoding cost weight ($0 < TCPOC_{depth_i} < 1$) and can be used to determine whether the CU depth selected by RDO is the best level.

With (1) and (7), it is readily seen that:

$$\frac{\frac{\lambda_{Mode}R_{nontexture,depth_i} + J_{RD_{texture,depth_i}}}{J_{RD_{texture,depth_i}}}}{< \frac{\lambda_{Mode}R_{nontexture,depth_{i+1}} + J_{RD_{texture,depth_{i+1}}}}{J_{RD_{texture,depth_{i+1}}}}} \tag{9}$$

From (9), a consistent relationship can be derived for $depth_i < depth_{i+1}$ and $0 < TCPOC_{depth_i} < 1$ as follows:

$$TCPOC_{depth_i} \geq TCPOC_{depth_{i+1}} \tag{10}$$

This relationship between *TCPOC* and *depth* can be verified in Figure 3, such as for the Class A sequence "Traffic". As shown in Figure 3, four quantization parameters "22, 27, 32, 37" were used in the experiment, and the *TCPOC* of all depth levels was obtained through actual coding. The *TCPOC* data corresponding to each quantization parameter were averaged. The more obvious the decrease in *TCPOC* between the levels, the deeper the depth level decreases; for example, the depth level "2" with the quantization parameter "22" is reduced from 0.888 to the depth level "3" 0.761. The monotonically decreasing relationship between *TCPOC* and depth level is shown.

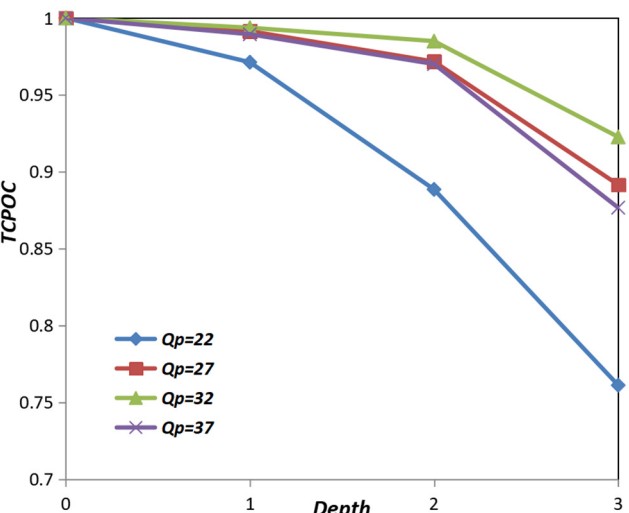

**Figure 3.** Relationship between *TCPOC* and depth for Class A sequence "Traffic" with different *Qp*.

### 3. Proposed Fast CU Size Decision Scheme

In Section 2, the non-texture and texture cost terminated threshold and the early skip CU detection were introduced. These concepts are used for a fast CU size decision algorithm. In fact, an effective prediction strategy is the best compromise between reducing computational complexity and maintaining rate-distortion performance in video coding. According to the analysis of Section 2, the distribution characteristics of the optimal depth structure of the coding tree unit have a high correlation with the video content, and there is a linear relationship between the non-texture cost and the texture cost in the depth of the coding unit. Therefore, these features are used to optimize the coding unit decision algorithm to obtain the optimal coding unit depth and to reduce the computational complexity of the video encoder. The algorithm combines the early termination classification model and the early skip model to determine the effective coding unit selection.

### 3.1. Early CU Termination Threshold Selection with TCPOC

As shown in Figure 1, the coding sequence of the CTU adopts the z-scan mode, and the coding is performed in the order of depth level. The numbers on the coded depth level indicate the coding order of each coding unit. This order is to ensure that each coding unit can obtain the reference CU above and on the left side of the CU during encoding. The reference information is the information of the coded unit and is used to predict the coding parameters of the current CU. Of course, the coding unit located at the top of the slice or the left border of the image frame needs to be removed.

To achieve the best RD–complexity trade-off in the mode decision process, the optimal threshold with the lowest process complexity for a given prediction accuracy should be chosen. The early determination strategy allows one to save some computation burden in the mode process by the *TCPOC* threshold with a descending order and to stop the process either because a better rate distortion performance cannot be arrived or because a minimal cost threshold is attained.

As discussed in Section 2.1, there is a one-to-one correspondence between the coding quadtree structure and the *TCPOC* value of a coding CU. Therefore, the best coding quadtree structure of the current CU is specified by the optimal *TCPOC*. If the best coding quadtree structure is reached before the final depth CU is processed, without any coding efficiency degradation, the remaining partitioning CUs can be skipped with less size. It should be pointed out that, on the one hand, the accuracy of intra-prediction increases with increasing the greater depth of the CTU, since the average prediction residual difference between a predicted sample and the reference sample decreases. On the other hand, the average efficiency of prediction coding typically increases with the smaller depth value. Hence, the optimal depth structure feature that allows the skipping of some partition units provides the possibility to select a suitable trade-off between the complexity reduction and coding efficiency.

Taking into account that the partitioning of the current CU depth into the next depth provides a more suitable description with accurate prediction parameters, in order to reduce the total cost, the next depth should be processed depending on the $TCPOC_{depth_i}$ of the current depth. When $TCPOC_{depth_i}$ is below a pre-set threshold TH corresponding to the optimal *RD* cost, the non-texture cost takes up a large portion of the total cost. Thus, no further cost reduction in the next depths is necessary. Based on the coupling relationship between *TCPOC* and depth level from level 0 to level 2 in each CU, an early determination scheme is designed to determine if the processing of the next depth can be skipped. An appropriate $TCPOC_{depth_i}$ threshold can accelerate the CU size decision scheme to the maximum extent while still maintaining a high coding efficiency.

### 3.2. Design of Early Termination Classification Model Based on Fuzzy Support Vector Machine

The rate-distortion improvement of the HEVC video encoder is mainly due to the high prediction accuracy of the CU partition. A smaller coding unit structure is used for high-texture complexity content, and a larger coding unit structure is used for smoother and low-texture complexity content. Therefore, the texture feature is the key feature that determines the size of the coding unit. In this paper, the spatial texture complexity, directional texture complexity, and coding unit texture content difference complexity are selected as texture features.

The content variance can accurately represent the spatial content complexity. Therefore, the spatial content complexity of each coding unit in this paper is defined as:

$$CT_{CU} = \frac{1}{N_{CU}} \sum_{(i,j) \in CU} \left[ f(i,j) - \frac{1}{N_{CU}} \sum_{(i,j) \in CU} f(i,j) \right]^2 \tag{11}$$

where $N_{CU}$ is the number of pixels of the current coding unit *CU*, and $f(i,j)$ is the brightness value of the $(i,j)$ pixel in the current coding unit. The variance can accurately describe the global content complexity of the coding unit, and the complexity of local details needs to

introduce other features. Intra-frame coding emphasizes the directionality of coding units and reference blocks. In this paper, the directional texture complexity is defined as:

$$DT_{CU} = \frac{1}{N_{CU}} \sum_{(i,j) \in CU} \left[ |S_{Hor}(i,j)| + |S_{Ver}(i,j)| + |S_{45°}(i,j)| + |S_{135°}(i,j)| \right] \qquad (12)$$

where $S_{Hor}(i,j)$, $S_{Ver}(i,j)$, $S_{45°}(i,j)$, and $S_{135°}(i,j)$ represent the $3 \times 3$ Sobel gradient in the directions of horizontal, vertical, and 45° and 135°, respectively.

The different depths of the coding units have their own content complexity, and the $16 \times 16$ size coding unit is used as the basic unit to extract the content texture difference between the depths of the coding unit. The difference complexity of the coding unit texture content can be expressed as:

$$DV_{CU} = \frac{1}{N_{sub\_CU}} \sum_{C_i \in CU} \left[ CT_{C_i} - \frac{1}{sub\_CU} \sum_{C_i \in CU} CT_{C_i} \right]^2 \qquad (13)$$

where $C_i$ is the $i$th sub-coding unit of the current $CU$ and $N_{sub\_CU}$ is the number of the smallest coding units in the current coding unit, $i = \{1, 2, 3, \ldots, 16\}$.

By combining the above texture content features, using the non-texture features including the current CTU segmentation structure information, the bit rate of the header, and the quantization step, etc., the problem of predicting the early termination of the CU size decision can be described as a binary classification problem. Many types of classifiers are used to solve binary classification problems. Support vector machines are machine learning classifiers based on statistical learning. In traditional SVM, all sample data have the same importance to the classification hyperplane, and the classifier assigns the same penalty factor to all the sample data. However, in the application of intra-frame coding unit classification, sample data are often affected by noise and feature distribution, and have different effects on the classification hyperplane. Therefore, in order to solve such practical problems, the fuzzy support vector machine introduces fuzzy membership degree, assigns different membership degrees to different sample data, distinguishes the different degrees of influence of the sample data on the classification hyperplane, and determines a more accurate classification hyperplane.

For a given training data set $\{(x_i, y_i), u(x_i)\}_{i=1}^n$, $x_i$ represents the feature vector of each sample, $x_i \in R^d$; $y_i$ represents two different classification of $y_i \in \{-1, +1\}$; $u(x_i)$ is a fuzzy membership function, indicating the reliability of the $i$-th sample $x_i$ belonging to the $y_i$ category, $0 < u(x_i) \leq 1$; and support vector machines use the feature mapping function $\phi(\cdot)$ to map the training samples to the high-dimensional feature space, namely $R^d \rightarrow R^F$, the converted training samples $\{\phi(x_i), y_i, u(x_i)\}$ are obtained, and the classification hyperplane is $\omega \times \phi(x_i) + b = 0$, where the kernel function is $K(x_i, x_j) = \phi(x_i)^T \phi(x_j)$. In intra-frame predictive coding, the depth distribution of the coding units is not uniform, and the sample distribution is unbalanced. As shown in Figure 4, for the distribution of the positive samples A and B, if the traditional distance membership degree is used to express the influence relationship, the two sample points have the same membership degree, but in fact the sample point B has similar negative sample distribution attributes. Therefore, the local distribution attributes of the samples have better classification characteristics. This paper proposes a membership function combining the distance scale and the local distribution scale of information entropy to more accurately solve the influence of noise and abnormal points.

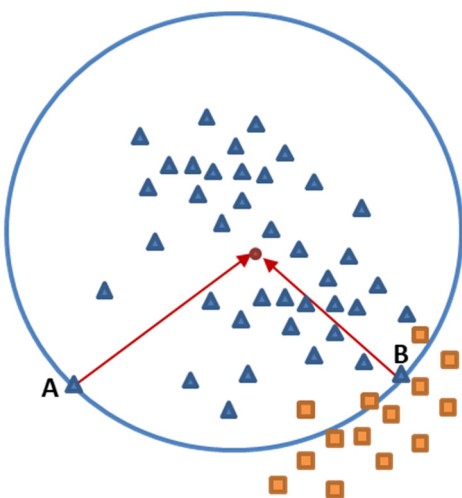

▲ Positive sample ■ Negative sample ● Positive Center

**Figure 4.** Schematic diagram of the relationship between sample distribution and membership.

Introducing the imbalance factor of the data set, the general form of the fuzzy support vector machine can be expressed as:

$$\min \frac{1}{2}\|\omega\|^2 + C^+ \sum_{\{i=1|y=+1\}}^{n} u_i^+ \xi_i + C^- \sum_{\{i=1|y=-1\}}^{n} u_i^- \xi_i$$
$$s.t.\ y_i[\omega \times \phi(x_i) + b] - 1 + \xi_i \geq 0, \xi_i \geq 0, i = 1, 2, \ldots n \tag{14}$$

In the formula, $C^+$ and $C^-$, respectively, represent the penalty factor of positive and negative sample data, and $\xi_i$ represents the relaxation factor. The optimal classification hyperplane of Equation (14) is solved by the Lagrangian multiplier method, as follows:

$$L(\omega, b, \xi, \alpha, \beta) = \frac{1}{2}\|\omega\|^2 + C^+ \sum_{\{i=1|y=+1\}}^{n} u_i^+ \xi_i + C^- \sum_{\{i=1|y=-1\}}^{n} u_i^- \xi_i$$
$$- \sum_{i=1}^{n} \alpha_i(y_i[\omega \times \phi(x_i) + b] - 1 + \xi_i) - \sum_{i=1}^{n} \beta_i \xi_i \tag{15}$$

According to the above formula, the dual plan is obtained:

$$\min \frac{1}{2}\sum_{i=1}^{n}\sum_{j=1}^{n} \alpha_i \alpha_j y_i y_j K(x_i, x_j) - \sum_{i=1}^{n} \alpha_i \tag{16}$$

The constraints are:

$$\sum_{i=1}^{n} \alpha_i y_i = 0, 0 \leq \alpha_i \leq u_i C^+, y_i = +1, i = 1, 2, \ldots, n^+$$
$$0 \leq \alpha_i \leq u_i C^-, y_i = -1, i = 1, 2, \ldots, n^- \tag{17}$$

The final decision function of the optimal classification hyperplane is:

$$f(x) = \text{sgn}(\sum_{i \in SV} \alpha_i y_i K(x_i, x) + b) \tag{18}$$

The centers of the positive and negative classes are:

$$\theta^+ = \frac{1}{m} \sum_{\{i=1|y=+1\}}^{m} x_i$$
$$\theta^- = \frac{1}{n-m} \sum_{\{i=1|y=-1\}}^{n-m} x_i \tag{19}$$

In the formula, the number of positive sample data is $m$ and the number of negative sample data is $n - m$.

$$R^+ = \max_{y_i = +1} |x_i - \theta^+|$$
$$R^- = \max_{y_i = -1} |x_i - \theta^-| \tag{20}$$

Substituting the above formula can get:

$$u_1(x_i) = \begin{cases} 1 - \frac{|x_i - x^+|}{R^+ + \delta}, & y_i = +1 \\ 1 - \frac{|x_i - x^-|}{R^- + \delta}, & y_i = -1 \end{cases} \tag{21}$$

where $\delta$ is a small positive number to avoid a denominator of 0 and to ensure that $0 \leq u_1(x_i) \leq 1$.

According to the concept of information entropy, the average amount of uncertainty information of $x_i$ belonging to the positive and negative category is as follows:

$$H^+(x_i) = -p^+(x_i) In(p^+(x_i))$$
$$H^-(x_i) = -p^-(x_i) In(p^-(x_i)) \tag{22}$$

where $p^+(x_i)$ and $p^-(x_i)$, respectively, represent the probability that $x_i$ belongs to the positive and negative class, and the probability is obtained by sampling through random neighbors. That is, the $k$ sample points $\{x_1, x_2, \ldots, x_k\}$ with the Euclidean distance closest from the current sample point $x_i$ are selected as the sampling data set, and the number of positive and negative samples in the data set $m^+$, $m^-$ can be calculated. The probability of the negative class is $p^+(x_i) = \frac{m^+}{k}$, $p^-(x_i) = \frac{m^-}{k}$. After getting the average amount of information of $x_i$, a new fuzzy membership function can be obtained:

$$u_2(x_i) = 1 - [H^+(x_i) + H^-(x_i)] \tag{23}$$

where $0 \leq u_2(x_i) \leq 1$.

Therefore, by fusing the two Equations (21) and (23), the fuzzy membership function of each sample data can be defined as:

$$u(x) = (1 - \eta)u_1(x_i) + \eta u_2(x_i) \tag{24}$$

where $\eta$ is the control factor, $0 < \eta < 1$, ensuring that $0 < u(x_i) \leq 1$.

In order to improve the classification accuracy of fuzzy support vector machines and to reduce the additional computational complexity of classification, the early termination of the classification model training proposed in this paper is divided into two stages: offline learning training and online fuzzy membership update. Offline learning training is using the original video reference encoder to encode multiple types of video sequences directly to generate a training data set. After obtaining a more accurate classification model, the model is loaded into the video encoder to process the coding unit to terminate the mode decision processing in advance; in the prediction stage, the first encoding frame of each video sequence uses the original video reference encoder to obtain accurate fuzzy membership parameters and to obtain more accurate imbalance factors, which are used to terminate the decision of the fuzzy support vector machine model early. In addition, in order to train the model to obtain a robust classification model, in two video sequences "BQTerrace (1920 × 1080)" and "BasketballDrill (832 × 480)", the first 50 frames of the video are chosen, using "22", "27", and "32", 37" four quantization parameters to generate training data sets. The two video sequences have different video content characteristics and spatial resolutions, which are suitable for obtaining complete training.

### 3.3. Initial Best Depth Prediction

Video content has a high degree of statistical correlation with spatial units. Therefore, the size of the coding unit is also highly correlated, and this spatial correlation has been used for coding unit prediction in many research works [6,7]. However, in these works,

only the CTU is the target of prediction, and the correlation between the inner code depths of the CTU and the correlation of the same spatially adjacent coding unit sizes have not been deeply explored. In this paper, we not only use spatial correlation features, but also introduce rate-distortion cost-related features between coding unit depths, design a more accurate fuzzy support vector machine classifier, and make decisions about coding unit splits in advance.

It can be seen from Table 1 that when the video content is smooth or the quantization parameter is large, the proportion of the best coding unit depth level of "0" is very high, reaching an average of 19.8%, while the video sequence "Kimono" is nearly 30% under the conditions of each quantization parameter. These statistical data also show that the best coded depth is directly predicted as "0" at the CTU level, and subsequent coding unit depth level calculations will be all saved. Therefore, in this paper, a fuzzy support vector machine classifier 0 is proposed in the stage of the depth level "0" stage to predict the result where the best depth level is "0". The main idea is to use the correlation between the rate-distortion cost of the current depth level "0" and the optimal rate-distortion cost of spatially adjacent CTUs, that is, the upper CTU, the left CTU, the upper left CTU, and the upper right CTU, combined with the average of the coded CTU to optimize the rate-distortion cost, form a candidate set, select the smallest rate-distortion cost, and use it to design the classifier.

In order to accurately describe the relationship between the optimal rate-distortion cost and the spatially adjacent CTUs, the normalized optimal rate-distortion cost difference rate is defined:

$$RDD = \frac{\left| RD_{opt\_cur} - RD_{opt\_min-ref} \right|}{RD_{opt\_cur}} \tag{25}$$

where $RD_{opt\_cur}$ is the optimal rate-distortion cost of the current CTU and $RD_{opt\_min-ref}$ is the smallest rate-distortion cost in the candidate set.

The original HM encoder is used to encode three standard video sequences (BQTerrace, BasketballDrill, Kimono) and to collect actual data to get Figure 5. It can be seen from the figure that there is a high degree of correlation between the current CTU and the rate-distortion cost of the coded CTU. Nearly 80% of the $RDD$ is less than 5%, and more than 90% of the $RDD$ is less than 10%. Therefore, the rate-distortion cost correlation has a high prediction stability.

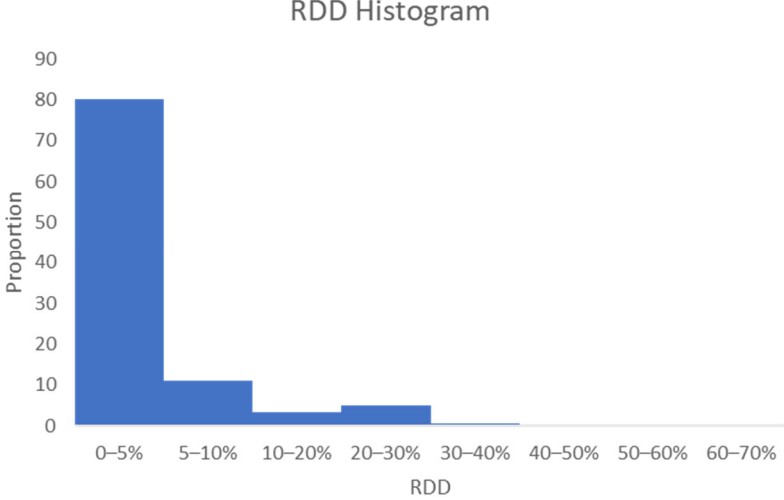

**Figure 5.** *RDD* Histogram.

If the classifier 0 determines that the depth level "0" is a non-optimal depth, then enter the classifier 1, as shown in Figure 6 In the design of classifier 1, based on the relevant information obtained by the actual rate-distortion optimization calculation for the coding unit depth "0", the relevant features between the depths are extracted to determine in

advance whether to skip the rate-distortion processing of the coding unit of the current depth level. As shown in Figure 7, directly using the information of the coded depth level "*i*", the rate distortion cost of the depth level "*i*" is:

$$RD_i = \frac{1}{N \cdot M} \sum_{i,j \in C_i} (o_{ij} - r_{ij})^2 + \lambda_{\text{mode}} R \qquad (26)$$

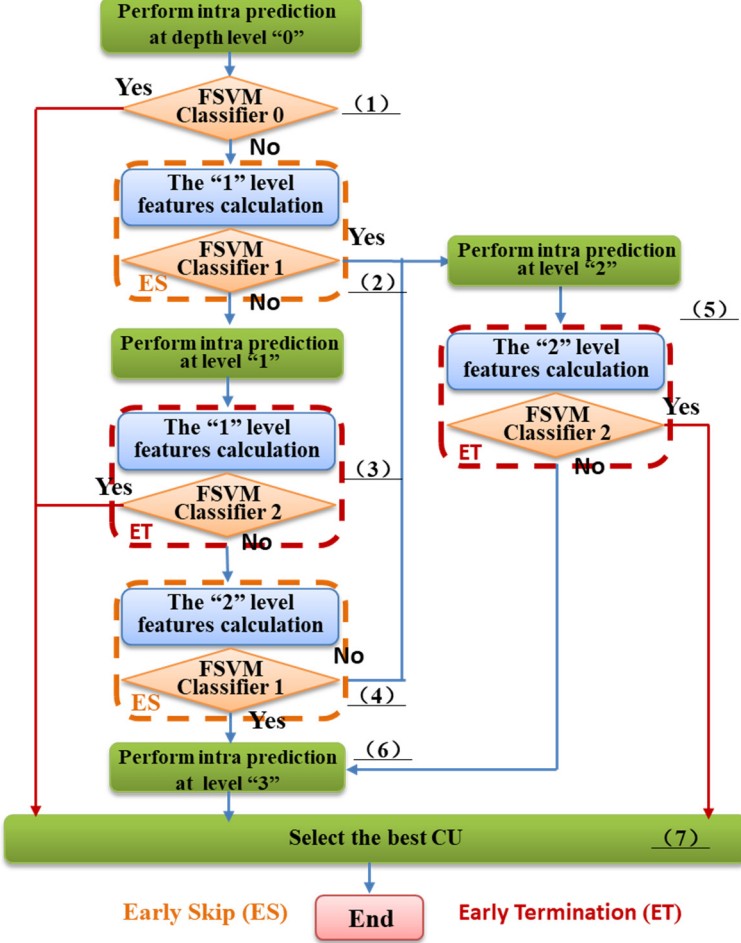

**Figure 6.** The flowchart of the proposed fast CU size decision algorithm.

According to Figure 6, the current coding unit depth level is further divided into $i + 1$, and the sub-region of the depth level $i$ corresponding to the coding unit of depth level $i + 1$ is defined as $(i, k)$, where $k \in \{0, 1, 2, 3\}$. The coded area is divided into four parts according to the sub-blocks of the corresponding area, and the classification is carried out according to the header information and residual information. The corresponding relationship is:

$$RD_i = \frac{1}{N \cdot M} \sum_{i,j \in C_i} D_{C_{i,j}} + \lambda_{\text{mode}} \left( R_{i,non-texture} + \sum_{i,j \in C_i} R_{C_{i,j},texture} \right) \qquad (27)$$

where $D_{C_{i,j}}$ is the coding distortion $D_{C_{i,j}} = \sum_{i,j \in C_{i,k}} (o_{ij} - r_{ij})^2$ of each sub-region corresponding to the depth level "*i*". Similarly, the corresponding texture part code rate is $R_{C_{i,j},texture}$ and the non-texture information obtained from the depth level "*i*" coding is $R_{i,non-texture}$, and the part that is equally distributed to each sub-region is $\frac{R_{i,non-texture}}{4}$. Then, Equation (27) becomes:

$$RD_i = \frac{1}{N \times M} \sum_{i,j \in C_i} D_{C_{i,j}} + \lambda_{\text{mode}} \sum_{i,j \in C_i} R_{C_{i,j}} = \sum_{i,j \in C_i} RD_{C_{i,j}} \qquad (28)$$

where

$$R_{C_{i,j}} = \frac{R_{i,non-texture}}{4} + R_{C_{i,j},texture}. \tag{29}$$

It can be seen from Section 2.2 that there is a high correlation between the minimum rate-distortion cost of spatially adjacent CTUs and the current CTU.

$$T_{RD_{opt,ref}} = \frac{\beta}{4}\min(RD_{opt,i}) \tag{30}$$

is used to evaluate the sub-block segmentation complexity of the current coding unit to determine whether to omit the current coding unit division jumps directly to the next depth level, where $RD_{opt,i}$ is the smallest CTU $RD$ cost in the candidate set and $\beta = 0.95$. The threshold calculation for skipping the segmentation process can be defined as:

$$SN_{pre} = \sum_{k=0}^{3} B(k) \tag{31}$$

where

$$B(k) = \begin{cases} 1 & if(RD_{C_{i,j}} > T_{RD_{opt,ref}}) \\ 0 & otherwise \end{cases}. \tag{32}$$

The original HM encoder is used to encode three standard video sequences (BQTerrace, BasketballDrill, and Kimono) with the $SN_{pre}$ value of three to and collect the actual data to get Table 4.

**Table 4.** The prediction accuracy over different sequences and $Qp$s.

| Video Sequence | $Qp$ | Accuracy (%) |
|---|---|---|
| BQTerrace (1920 × 1080) | 22 | 93.4 |
| | 27 | 92.5 |
| | 32 | 92.8 |
| | 37 | 93.6 |
| BasketballDrill (832 × 480) | 22 | 96.1 |
| | 27 | 95.5 |
| | 32 | 95.8 |
| | 37 | 96.2 |
| Kimono (1920 × 1080) | 22 | 98.3 |
| | 27 | 97.8 |
| | 32 | 97.6 |
| | 37 | 98.4 |
| Average | | 95.67 |

It can be seen from Table 4 that for different quantization parameters $Qp$, the accuracy rate is above 92%, the average accuracy rate is 95.67%, the accuracy rate is very high, and the accuracy rate is also very stable under different quantization parameters. Especially for video sequences (BQTerrace, BasketballDrill), most of the optimal depth levels are greater than "2", and the accurate prediction results can skip the rate distortion calculation of depth levels "1" and "2". Therefore, the rate-distortion cost correlation decision between depth levels also has higher prediction stability.

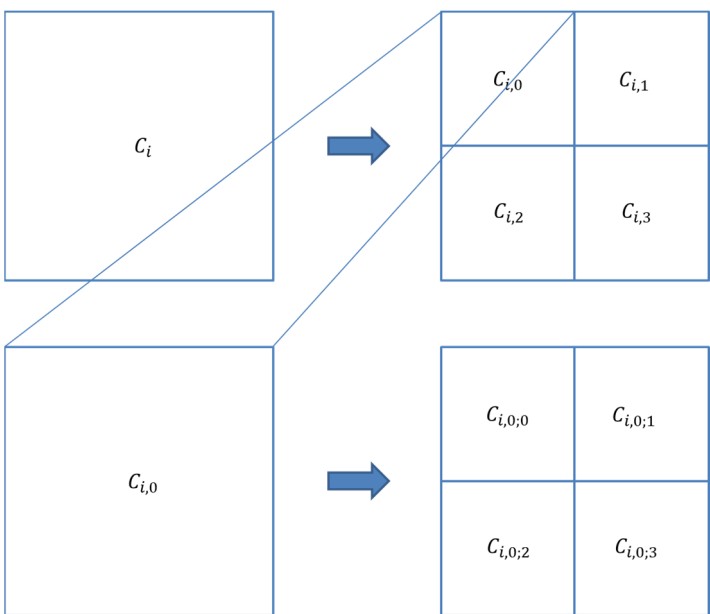

**Figure 7.** The partition of the "0" depth level and its corresponding sub-CU areas.

### 3.4. Overall Algorithm

As shown in Figure 6, the fast optimization decision-making scheme of coding unit size proposed in this paper includes three fast algorithms: the optimal coded depth level "0" early decision, coding depth level skip early, and coding unit size termination early. The three algorithms proposed in this paper are designed based on the texture and non-texture rate-distortion relationship so that each unit can make an optimal decision based on the space and the internal rate-distortion cost of the CTU. In addition to the encoded features, space complexity features and non-texture/texture relationship features are also applied to our fast algorithm. In addition, for the characteristics and statistical distribution of the coding unit size, a set of fuzzy support vector machine classifiers are designed, which are used in three fast algorithms to obtain adaptive feature thresholds under different trade-off requirements so that we can effectively balance *RD* performance and complexity. For each input video sequence, the first three frames of images are encoded by the original HM encoder, and the actual rate-distortion optimized coding unit size results and the corresponding feature data are obtained, which are used to train the current FSVM to classify 0, 1, and 2, respectively.

According to the above analysis process, our proposed coding unit size decision algorithm is summarized as follows:

Step (1): Start encoding from the depth level of "0", perform rate-distortion optimization encoding processing, and obtain the code rate and distortion cost of the depth level of "0". Based on the encoded information and spatially related coding unit information, collect the feature vectors related to the classifier, and input the collected feature vector data into the classifier "0" to obtain the discrimination result. If it is "Yes", it will exit the subsequent encoding depth as the depth level "0" is the best encoding size; otherwise, if it is "No", it will continue the subsequent depth level encoding and will go to step (2).

Step (2): On the basis that the depth level "0" has been encoded, enter the early skip classifier to determine whether to enter or omit the encoding calculation of the depth level "1". Using the partition structure shown in Figure 7, the rate-distortion cost of the corresponding sub-coding unit is extracted, and the feature vector obtained in step (1) is combined into the classifier "1". If the classification result is "Yes", the current depth level is directly omitted, and the processing of the depth level "2" is performed, and then it will go to step (5); otherwise, the classification result is "No", and the rate distortion calculation of the current depth level "1" is directly entered. Go to step (3).

Step (3): Execute the rate-distortion optimization process of the depth level "1" coding unit, obtain the feature vector data of the current depth level, and input the early termination classifier 2. If the early termination is "Yes", then the depth level "1" is the optimal coding depth; otherwise, go to step (4).

Step (4): Perform depth level "1" early skip classifier processing and determine whether to enter or omit the depth level "2" encoding calculation. Using the partition structure shown in Figure 7, the rate-distortion cost of the sub-coding unit corresponding to the depth level "1" is extracted, the corresponding feature vector is extracted, and it is inputted to the classifier "1". If the classification result is "Yes", directly omit the coding calculation of the depth level "2" and go to the processing step (6) of the depth level "3"; otherwise, enter the rate distortion calculation of the depth level "2" and go to step (5).

Step (5): Execute the rate-distortion optimization process of the depth level "2", obtain the feature vector data of the current depth level, and input the early termination classifier 2. If the early termination is "Yes", then the depth level "2" is the optimal coding depth; otherwise, go to step (6).

Step (6): Execute the rate-distortion calculation of the depth level "3" and go to step (7).

Step (7): Choose the best coding depth level.

## 4. Experimental Results

In this section, the experiment was carried out in All-Intra-mode under the HM16.5 reference software model [26]. Four quantization parameters 22, 27, 32, 37 were chosen to encode 18 video sequences in the JCT-TC standard test set with five classes [27] (A (4K×2K), B (1080p), C (WVGA), D (QWVGA), and E (720p)). All experiments were performed on a computer equipped with Intel Core i7-6700K CPU @4.0 Hz (Intel, Santa Clara, CA, USA), 32 G memory, and a Windows 10 64-bit operating system (Microsoft Corporation, Redmond, WA, USA). The original HEVC software test model (HM16.5) served as a reference anchor point for performance comparison. The reduction in the computational complexity was measured by TS. The calculation of TS on the original HM was obtained according to formula (33):

$$\text{TS} = \frac{T_{pro} - T_{org}}{T_{org}} \times 100\% \tag{33}$$

where $T_{org}$ and $T_{pro}$ are the encoding time of the original method in HM16.5 and the proposed fast scheme, respectively.

The *RD* performance gain was evaluated by BDBR [28]. A larger value of TS means more computational complexity was reduced, and a positive value of BDBR indicates performance degradation. For the proposed method, the encoding time is the total encoding execution time, including the training time of all classifiers. For each video sequence, the first three frames were selected as the training data of the classifier online training mode for classifier training. After the classifiers were prepared, it was directly used for the CTU prediction decision of the subsequent frames.

The test experimental data of the three-level classifiers involved in this article are shown in Table 5, comparing the *RD* performance and TS of the reference software HM. First of all, classifier 0 was the optimal depth level 0 to stop the judgment. This method achieved an average 22% reduction in computational complexity and maintained an average loss of 0.4% BDBR; from the experimental data, it can be seen that the classifier accurately separated the coding unit with the optimal depth level of "0", especially the video sequences "Traffic", "ParkScence", "FourPeople", and "Johnny", and the proportion of the optimal depth level of "0" was relatively large. Secondly, classifier 3 was the optimal coding depth level to terminate the decision early. This solution achieved the largest reduction in the calculation range, reducing the complexity by an average of 40%, while maintaining a low BDBR, which was 0.36%; from the experimental data, it can be seen that this part was the main contribution to the reduction in computational complexity, especially for video sequences with low content complexity, such as "Cascus", "Kimono", "PartyScene", and "Johnny". The optimal depth level of this type of video was that the proportion of C5 and C6 was

small, and the classifier could accurately determine the optimal depth to stop and exit in time. Finally, classifier 2 was a non-optimal coded depth skipping judgment; excluding the above two types of classifiers, the optimal depth level in the coding unit tree was not high. Therefore, this method achieved an average of 26% of the computational complexity and the BDBR rate-distortion performance was reduced by an average of 0.49%.

**Table 5.** Results of two individual schemes compared to HM16.5 encoder.

| Video Type | Video Sequences | Classifier 1 | | Classifier 0 | | Classifier 2 | |
|---|---|---|---|---|---|---|---|
| | | BDBR (%) | TS (%) | BDBR (%) | TS (%) | BDBR (%) | TS (%) |
| A | PeopleonStreet | 0.78 | −25 | 0.78 | −34 | 0.18 | −35 |
| | Traffic | 0.36 | −34 | 0.36 | −26 | 0.08 | −41 |
| B | BQTerrace | 0.58 | −23 | 0.58 | −30 | 0.09 | −23 |
| | Cacus | 0.52 | −28 | 0.52 | −22 | 0.21 | −45 |
| | Kimono | 0.41 | −21 | 0.41 | −29 | 0.26 | −50 |
| | ParkScene | 0.34 | −34 | 0.34 | −32 | 0.31 | −44 |
| C | BasketballDrill | 0.65 | −26 | 0.65 | −16 | 0.16 | −42 |
| | BQMall | 0.37 | −22 | 0.37 | −29 | 0.2 | −40 |
| | PartyScene | 0.26 | −18 | 0.26 | −21 | 0.81 | −55 |
| | RaceHorses | 0.21 | −13 | 0.21 | −27 | 0.58 | −31 |
| D | BasketballPass | 0.24 | −17 | 0.24 | −22 | 0.61 | −40 |
| | BlowingBubbles | 0.11 | −11 | 0.11 | −26 | 0.25 | −41 |
| | Bqsquare | 0.21 | −15 | 0.21 | −27 | 0.35 | −30 |
| | RaceHorses | 0.09 | −9 | 0.09 | −30 | 0.2 | −35 |
| E | FourPeople | 0.41 | −34 | 0.41 | −24 | 0.2 | −43 |
| | Johnny | 0.91 | −35 | 0.91 | −27 | 1.2 | −49 |
| Average | | 0.40 | −22 | 0.49 | −26 | 0.36 | −40 |

In order to demonstrate the superiority of our proposed approach, four state-of-the-art fast HEVC encoding algorithms, including Jamali TB [13], Liu TB [19], Xu TIP [21], and Yan CSVT [23], were utilized as benchmarks for performance comparison. Among them, Jamali TB [13] is based on statistical information, Liu TB [19] is based on traditional machine learning, and Xu TIP [21] and Yan CSVT [23] are based on deep learning. The classifier was proposed to use only the actual coded data of the first three frames of the video sequence for training, and did not update. The specific experimental results are shown in Table 6. For Jamali TB [13], the average computational complexity was reduced by 46.75%, the *RD* performance lost 1.3% of BDBR, and Xu TIP [21] can lose BDBR 2.05%. In the case of Xu TIP [21], the computational complexity was reduced by 60.13% on average. Liu TB [19] maintained a good *RD* performance, but the reduction in computational complexity was limited. Yan CSVT [23] reduced the computational complexity the most, achieving a 61.5% reduction in computational complexity, and maintained BDBR's *RD* performance of 1.9%. For the proposed approach, it achieved a 53.24% reduction in computational complexity and the increased BRBR was only 0.82%. While maintaining the best *RD* performance, it achieved a very high computational complexity reduction. The proposed approach adopts a more efficient fuzzy support vector machine classifier design and introduces more feature vectors and unbalanced data adjustment factors, which solves the problem of outlier recognition and reduces the loss cost caused by misclassification. The experimental results showed that the proposed approach is superior to several other methods in terms of reducing complexity and *RD* performance integration, and obtained a higher computational complexity reduction while keeping the *RD* performance basically unchanged. At the same time, the standard deviation value of each scheme was calculated according to the *RD* performance loss and TS. The experimental data showed that the proposed approach had a relatively stable *RD* performance, while the proposed approach maintained a small amount of computational complexity to reduce fluctuations. Therefore, compared with these

algorithms, the proposed approach has the best rate-distortion-complexity performance. Compared with the fast CU decision method in [13], the proposed scheme performed better for most test sequences and achieved more than 6.49% coding time saving and a 0.48% BDBR decrease on average. Although the proposed scheme showed a slight increase in BDBR for some test sequences in class A and C, it can be balanced out to a considerable extent by the corresponding coding time reduction. As a whole, our proposed fast intra-mode decision scheme gives a better comprehensive performance by providing a trade-off between the rate distortion performance and the encoding complexity.

**Table 6.** Results of the proposed algorithm compared with a state-of-the-art fast algorithm.

| Video Type | Video Sequences | Proposed Algorithms | | Jamali TB [13] | | Liu TB [19] | | Xu TIP [21] | | Yan CSVT [23] | |
|---|---|---|---|---|---|---|---|---|---|---|---|
| | | BDBR (%) | TS (%) | BDBR (%) | TS (%) | BDBR (%) | TS (%) | BDBR (%) | TS (%) | BDBR (%) | TS (%) |
| A | PeopleonStreet | 1.32 | −53.2 | 1.71 | −49.4 | 1.39 | −59.2 | 2.37 | −61.0 | 1.89 | −61.3 |
| | Traffic | 1.25 | −64.5 | 1.46 | −48.8 | 1.40 | −61.3 | 2.55 | −70.8 | 1.74 | −63.1 |
| B | BQTerrace | 1.01 | −52.6 | 0.82 | −46.7 | 1.40 | −57.3 | 1.84 | −64.7 | 1.37 | −62.3 |
| | Cacus | 0.34 | −58.2 | 1.46 | −47.7 | 0.99 | −57.8 | 2.27 | −61.0 | 1.70 | −63.9 |
| | Kimono | 0.90 | −64.5 | 1.54 | −49.5 | 0.87 | −65.8 | 2.59 | −83.5 | 0.85 | −69.0 |
| | ParkScene | 0.52 | −52.4 | 1.02 | −47.4 | 1.08 | −57.3 | 1.96 | −67.5 | 1.70 | −63.6 |
| C | BasketballDrill | 0.32 | −48.1 | 0.85 | −48.7 | 1.50 | −57.4 | 2.86 | −53.0 | 3.48 | −63.8 |
| | BQMall | 1.24 | −45.7 | 1.48 | −47.1 | 1.29 | −54.9 | 2.09 | −58.4 | 2.24 | −62.3 |
| | PartyScene | 1.10 | −62.3 | 1.02 | −41.1 | 0.71 | −48.4 | 0.66 | −44.5 | 1.70 | −56.0 |
| | RaceHorses | 0.87 | −45.9 | 0.65 | −44.6 | 1.30 | −55.9 | 1.97 | −57.1 | 1.45 | −62.4 |
| D | BasketballPass | 0.75 | −51.2 | 1.22 | −46.5 | 0.82 | −56.7 | 1.84 | −56.4 | 2.09 | −62.9 |
| | BlowingBubbles | 0.34 | −56.9 | 1.03 | −44.2 | 0.47 | −53.6 | 0.62 | −40.5 | 2.05 | −56.0 |
| | Bqsquare | 0.41 | −44.5 | 1.29 | −41.0 | 0.32 | −59.1 | 0.91 | −45.8 | 1.50 | −47.9 |
| | RaceHorses | 0.32 | −43.1 | 1.22 | −46.5 | 0.51 | −52.3 | 1.32 | −55.8 | 1.65 | −57.7 |
| E | FourPeople | 0.60 | −55.3 | 1.78 | −48.9 | 1.66 | −62.0 | 3.11 | −71.3 | 2.30 | −62.8 |
| | Johnny | 1.90 | −53.4 | 2.22 | −49.9 | 2.16 | −69.6 | 3.82 | −70.7 | 2.61 | −69.0 |
| | Average | 0.82 | −53.24 | 1.30 | −46.75 | 1.12 | −58.04 | 2.05 | −60.13 | 1.90 | −61.50 |

## 5. Conclusions

In this paper, a multi-level fast CU size decision algorithm based on machine learning is proposed. The CU depth level is predicted by the fuzzy SVM classifier, and the multi-level combined different classification task of the CTU partition quad-tree structure is constructed to achieve the best compromise between coding computational complexity and rate distortion. A trade-off between performance in favor of the large-scale implementation of HEVC applications and the software encoding implementation of mobile real-time video applications was established. Different from the traditional SVM classifier design, the coding unit decision-making scheme based on the kernel fuzzy support vector machines in this paper can adaptively solve the problem of classification prediction accuracy. Then, using the feature of the unbalanced data of the coding unit, a membership function combining the distance scale and the local distribution scale of information entropy is introduced to accurately eliminate the negative influence of data noise and abnormal values. Finally, in order to obtain a better rate-distortion complexity performance, different classification tasks are applied at different stages of depth-level "0", depth-level skip prediction, and early termination, and direct prediction is performed through the kernel fuzzy support vector machine classifier. The experimental results showed that compared with the original video coding test model HM16.5, the proposed scheme successfully reduces the coding time by about 53.24% on average, while effectively maintaining an almost unchanged coding performance. Compared with other existing statistical information and machine learning

fast CU size decision algorithms, the proposed method can achieve better rate-distortion complexity performance.

**Author Contributions:** Conceptualization and methodology, S.H. and C.S.; software, Z.D.; validation, S.H., C.S. and Z.D.; formal analysis, Z.D.; investigation, S.H.; data curation, C.S. and Z.D.; writing—original draft preparation, S.H. and C.S.; writing—review and editing, S.H. and Z.D.; visualization, Z.D.; supervision, C.S. All authors have read and agreed to the published version of the manuscript.

**Funding:** This research was funded by the Hainan Provincial Natural Science Foundation of China under Grant (No. 2019RC199). This work was Supported by the Hainan Province Key R&D Program Project (No. ZDYF2019010, ZDYF2021GXJS010). This work was Supported by the National Natural Science Foundation of China (No. 61562023, 61362016, 61502127). This work was supported by the Major Science and Technology Project of Haikou City (No. 2020006).

**Institutional Review Board Statement:** Not applicable.

**Informed Consent Statement:** Not applicable.

**Data Availability Statement:** Not applicable.

**Conflicts of Interest:** The authors declare no conflict of interest.

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
