# Peer review of "Fast Decision Algorithm of CU Size for HEVC Intra-Prediction Based on a Kernel Fuzzy SVM Classifier"

_electronics, doi:10.3390/electronics11172791_

Round 1
Reviewer 1 Report
The paper is interesting and timely.
Is the overall algorithm shown in Figure 7 always can exit from the loop formed by "FVSM Classification 2" and "Next depth level"?
Is recommended to add an explicit exit from this loop or comment on this.
The steps described below Figure 7 are difficult to link to the content of figure 7. Is recommended to add some notes in figure 7.
What is the source of video sequences used in the paper?
What is the source of video sequences used in the paper?
Editorial notes:
- in the paper, some editorial errors are noticed
- change Fig. to Figure - line 196
Author Response
1)Is the overall algorithm shown in Figure 7 always can exit from the loop formed by "FVSM Classification 2" and "Next depth level"?
2)Is recommended to add an explicit exit from this loop or comment on this.
3)The steps described below Figure 7 are difficult to link to the content of figure 7. Is recommended to add some notes in figure 7.
Reply: For the above three questions, Thanks to the reviewers for reminding that some content missing in Figure 7 is not clearly described. According to the reviewer's comments, Figure 7 was redrawn, specific steps were added to the figure, and a detailed description of the process in Figure 7 was re-added according to the new content of Figure 7.-line 609-671.
4)What is the source of video sequences used in the paper?
Reply:Sixteen sequences recommended by JCT-VC with five resolutions [28] are used in the paper.
5)
Editorial notes:
- in the paper, some editorial errors are noticed
- change Fig. to Figure - line 196
Reply:There are some formatting problems in the paper, which have been revised according to comments, such as line 54, 196, 263, 301.
All modifications are marked in red font.
Reviewer 2 Report
A multi-level fast CU size decision algorithm based on machine learning in order to achieve the trade-off between encoding computational complexity and rate-distortion performance was proposed in the submitted manuscript. The paper is well-written and interesting for readers, therefore it may be published after the minor revision as follow:
1- Revise the abstract to show the findings and novelty of your study.
2- Please refer to all figures in the context of the manuscript. Moreover, be consistent with Fig. and Figure as well as Font of context.
3- Please change the type of equation in the context to be at the same size and align with the text. For example see lines 292, 297, etc.
4- Please add a legend to Figure 3.
5- Line 305, there is only one Section 2, revise the sentence as "In section 2,".
6- It is better to write the manuscript in the third person language.
7- The quality of figure 6 is low. Please revise it.
8- The font of the manuscript from 524 to 582.
9- Please revise the conclusion to show the novelty as well as the objective of the study.
Author Response
1- Revise the abstract to show the findings and novelty of your study.
Reply:It has been modified as requested, adding innovations and new discoveries.
2- Please refer to all figures in the context of the manuscript. Moreover, be consistent with Fig. and Figure as well as Font of context.
Reply:There are some formatting problems in the paper, which have been revised according to comments, such as line 54, 196, 263, 301.
3- Please change the type of equation in the context to be at the same size and align with the text. For example see lines 292, 297, etc.
Reply:The full text equation type has been resized and re-arranged.
4- Please add a legend to Figure 3.
Reply:Resupplemented Figure 3 information and added legend-line-300-307.
5- Line 305, there is only one Section 2, revise the sentence as "In section 2,".
Reply:It has been modified according to the modification suggestion, thank you!
6- It is better to write the manuscript in the third person language.
Reply: The first person in the full text has been changed to the third person language, such as Line-234, 245, 287, 297, 313, 467, 486, 525.534, 539, 574, 582, 602.
7- The quality of figure 6 is low. Please revise it.
Reply:It has been replaced according to the revised comments, thank you!
8- The font of the manuscript from 524 to 582.
Reply:It has been modified according to the modification suggestion, thank you!Line-533-592.
9- Please revise the conclusion to show the novelty as well as the objective of the study.
Reply:The innovation points and research objectives have been supplemented according to the revised comments. Such as Line 750-755, 759-761.
All the above modifications have been marked in red font.